# High-harmonic generation from a flat liquid-sheet plasma mirror

Yang Hwan Kim[1], Hyeon Kim[1,2], Seong Cheol Park[1,2], Yongjin Kwon[1,2], Kyunghoon Yeom [1,2], Wosik Cho[1], Taeyong Kwon[1,2], Hyeok Yun[3], Jae Hee Sung[1,3], Seong Ku Lee [1,3], Tran Trung Luu [4], Chang Hee Nam [1,2] & Kyung Taec Kim [1,2] ✉

High-harmonic radiation can be generated when an ultra-intense laser beam is reflected from an over-dense plasma, known as a plasma mirror. It is considered a promising technique for generating intense attosecond pulses in the extreme ultraviolet and X-ray wavelength ranges. However, a solid target used for the formation of the over-dense plasma is completely damaged by the interaction. Thus, it is challenging to use a solid target for applications such as time-resolved studies and attosecond streaking experiments that require a large amount of data. Here we demonstrate that high-harmonic radiation can be continuously generated from a liquid plasma mirror in both the coherent wake emission and relativistic oscillating mirror regimes. These results will pave the way for the development of bright, stable, and high-repetition-rate attosecond light sources, which can greatly benefit the study of ultrafast laser-matter interactions.

When an intense femtosecond laser pulse is irradiated to the surface of a dense material, such as a solid or liquid, a thin layer of plasma can be created on the surface of the material. If the plasma is sufficiently dense, the laser field that arrives afterwards can be reflected. This phenomenon is known as plasma mirror[1,2]. The oscillatory motion of electrons in the plasma, driven by the laser field, is responsible for this reflection. When the laser intensity ($I$) is very high, the electron motion becomes highly nonlinear ($I > \sim 10^{15}$ W/cm$^2$ at the wavelength of 800 nm) or even relativistic ($I > \sim 10^{18}$ W/cm$^2$ at the wavelength of 800 nm), leading to the emission of high-energy photons in the specular direction. Two major emission mechanisms were identified[3]. One is coherent wake emission (CWE), which is generated by the coherent excitation of plasma wakes on the surface of the plasma mirror[4–8]. The other is relativistic oscillating mirror (ROM), in which the collective oscillation of the plasma surface induces a relativistic Doppler frequency upshift of the reflected laser pulse[9–12]. As these emissions occur periodically in the oscillating laser field, their spectrum exhibits a high-harmonic structure.

High-harmonic generation (HHG) on a plasma mirror is particularly interesting as it is considered a promising technique for generating intense attosecond pulses in the extreme ultraviolet (EUV) and X-ray wavelength ranges[4,9,13–15]. The attosecond pulses generated from a plasma mirror are known to possess good spatiotemporal coherence[16]. The conversion efficiency is higher than that of HHG in gases[3]. Furthermore, there is no upper limit on the peak intensity of a driving laser pulse, unlike HHG in gases[13,17–21]. Thus, intense coherent attosecond pulses can be generated through HHG from a plasma mirror, which is highly beneficial in applications for studying ultrafast laser-matter interactions.

HHG from a plasma mirror is also important for fundamental research. Recently, it was proposed that an ultrahigh intensity (~$10^{26}$ W/cm$^2$) can be achieved by utilising the attosecond pulses generated on the curved surface of a relativistic oscillating plasma mirror[11,22–24], which can enable the experimental investigation of strong-field quantum electrodynamic effects such as Breit–Wheeler pair production[24]. Thus, HHG from a plasma

[1]Center for Relativistic Laser Science, Institute for Basic Science, Gwangju 61005, Republic of Korea. [2]Department of Physics and Photon Science, Gwangju Institute of Science and Technology, Gwangju 61005, Republic of Korea. [3]Advanced Photonics Research Institute, Gwangju Institute of Science and Technology, Gwangju 61005, Republic of Korea. [4]Department of Physics, The University of Hong Kong, SAR Hong Kong, China. ✉e-mail: kyungtaec@gist.ac.kr

mirror is of great interest for both applications and fundamental research.

Despite its great potential, experimental studies related to HHG from a plasma mirror have been limited mainly to revealing its generation mechanisms[3,4,9,25] and plasma dynamics[22,26,27]. One of the fundamental problems in studying HHG and other laser-matter interactions in a plasma mirror is the destruction of the target in each interaction with an intense laser beam[28]. Therefore, most studies have been conducted in single-shot-based or low-repetition-rate (~0.1 Hz) experiments. It is difficult to use a plasma mirror in applications that require high flux or high repetition rate. Consequently, research related to HHG[16,22,29–31] from a plasma mirror[27] has progressed slowly compared with that of HHG sources based on gas targets operating at high repetition rates.

Efforts have been made to address this problem. Mechanical solutions such as rotating discs[32–34] and tape targets[6,35–37] have been successfully implemented in some experiments. The attosecond lighthouse effect on the surface of a plasma mirror[38] and control of the sub-cycle dynamics of an electron[26] have been demonstrated using a rotating fused silica disk. Recently, HHG in the relativistic regime at a repetition rate of 1 kHz has been demonstrated using the rotating disk method[39], and a detailed study of the plasma scale length was reported. HHG and other high-power laser experiments were performed using a tape target[40–42]. However, these solutions provide a limited number of laser shots. It is difficult to apply these approaches to applications such as attosecond streaking experiments or other pump-probe experiments that require millions or even more laser shots. Thus, a target that can provide unlimited laser shots at a high repetition rate is required.

The laminar flow of a liquid jet has been demonstrated as a promising target for high-repetition-rate experiments because the target is refreshed with its flow[43–56]. HHG from a liquid plasma was performed only once using a cylindrical liquid jet in a single-shot-based experiment[57]. In their experiment, HHG was observed in the CWE regime. HHG in the ROM regime was not observed due to the limited spectral range of the spectrometer, even though the peak intensity (~$10^{19}$ W/cm$^2$) was sufficiently high for HHG in the ROM regime. The cylindrical surface of the liquid jet caused issues with the reflected beam. A small deviation of the alignment causes a large deflection. Also, the HHG beam was divergent in one direction. These issues can be resolved by using a liquid flat-jet. Recently, it was reported that an ethylene glycol flat-jet with low vapour pressure can be used for laser-plasma experiments such as proton acceleration[47] and demonstration of the radial Weibel instability[51] in the relativistic regime at a repetition rate of 1 kHz. The existing research strongly suggests that a liquid flat-jet target with low vapour pressure can be used as a plasma mirror.

In this work, we demonstrate HHG from a liquid plasma mirror in both the CWE and ROM regimes. A liquid plasma mirror was prepared using a liquid flat-jet. We first demonstrate HHG from a liquid plasma mirror in the CWE regime at a repetition rate of 1 kHz. We investigate the properties of HHG from the liquid plasma mirror by changing the polarisation angle, ellipticity, group-delay dispersion (GDD) and peak intensity of the driving laser pulse. We confirm that the CWE harmonics could be generated continuously at a repetition rate of 1 kHz. We also demonstrate HHG from a liquid plasma mirror in the ROM regime using a 150-TW-laser that is equipped with double plasma mirrors for high temporal contrast[58]. We observed a ROM spectrum up to 33$\omega_0$ (51.1 eV, $\omega_0$ = 1.55 eV for a photon with a wavelength of 800 nm), which is significantly higher than the CWE cutoff of 14.1$\omega_0$ (21.8 eV), estimated for a fully ionised propylene glycol plasma. To the best of our knowledge, this is the first demonstration of HHG in both the CWE and ROM regimes using a liquid plasma mirror with continuous operation.

## Results

### Liquid flat-jet and coherent wake emission from liquid plasma mirrors

For the experimental demonstration of HHG from a liquid plasma mirror, we used the experimental setup illustrated in Fig. 1a. A propylene glycol flat-jet target was made by obliquely colliding two liquid jets. The two jets were made using capillary tubes. The propylene glycol flat-jet was mounted on a set of motorised stages consisting of motorised three-axis translation stages ($x$, $y$ and $z$), a motorised goniometer ($\theta$) and a motorised rotary stage ($\phi$). Thus, the positions and angles of the liquid-sheet target can be accurately controlled. This setup was used to control the reflection direction of the high-harmonic radiation generated along the specular direction.

We chose propylene glycol because the absorption of the harmonic radiation in the EUV wavelength range can be minimised due to its low vapour pressure (0.1 mbar). It also helps maintain a low pressure in the vacuum chambers during the experiments and minimises the contribution of HHG, if any, from the gas phase to the detected signal. The background pressures were maintained below $1.0 \times 10^{-2}$ and $3.0 \times 10^{-5}$ mbar in the target and spectrometer chambers, respectively, during the experiments. Note that we did not use a cold trap for the liquid catcher. Instead, we directly removed the liquid collected in the catcher using peristaltic pumps. The vacuum level and the shape of the liquid sheet target were stable during the operation. With this experiment setup, the liquid can be circulated. Thus, the experimental setup can be operated continuously even for several days long (see Figs. S1–S2 for a complete connection diagram in the Supplementary Information).

A flat sheet of propylene glycol was formed when the two liquid jets collided (Fig. 1b). The laser pulses were tightly focused using an off-axis parabolic mirror (f/2.5) on a liquid sheet with a repetition rate of 1 kHz at an incidence angle of 45°. Bright fluorescence at the focal spot (Fig. 1c) and purple fluorescence from the liquid plasma (Fig. 1d) were observed, where the plasma expanded along the direction normal to the liquid surface.

The reflected laser beam was sent to a flat-field EUV spectrometer. A typical angle-resolved EUV spectrum and its angle-integrated spectrum are shown in Fig. 1e, f, respectively. As the laser beam was tightly focused (f/2.5), the divergence of the high-harmonic radiation was large (>25 mrad). Both odd and even harmonics were observed with slowly varying intensities up to 22 eV (Fig. 1e, f), indicating that harmonic radiation was generated through the interaction between the laser field and the plasma surface. The peak intensity of the laser field on the liquid target was estimated as $2 \times 10^{16}$ W/cm$^2$. At this intensity, harmonic radiation is generated mainly through the CWE mechanism[3,21]. The maximum harmonic order also corresponds to the CWE cutoff of 14.1$\omega_0$ (21.8 eV), which was estimated from the plasma oscillation frequency of the fully ionised propylene glycol plasma. Harmonic radiation was produced continuously at a repetition rate of 1 kHz. The harmonic spectrum (Fig. 1e) shows a smooth intensity distribution in $\theta_y$, indicating that the surface quality of the liquid target is reasonably good on the scale of focal spot size of the driving laser field.

### Polarisation dependence of CWE from liquid plasma mirrors

The dependence of HHG from the liquid plasma mirror on the polarisation angle of the laser field was investigated. The polarisation angle of the linearly polarised laser pulse was controlled using a half-wave plate. The EUV spectra obtained at each polarisation angle were recorded, as shown in Fig. 2a. The laser pulses were p-polarised at $\alpha = 0°$ and s-polarised at $\alpha = 90°$. The harmonic intensity was maximised for the p-polarisation and minimised for the s-polarisation of the driving laser field. We performed 2D particle-in-cell (PIC) simulations (see "Methods" section) for various polarisation angles, $\alpha$. The PIC simulation results effectively reproduced the experimental result,

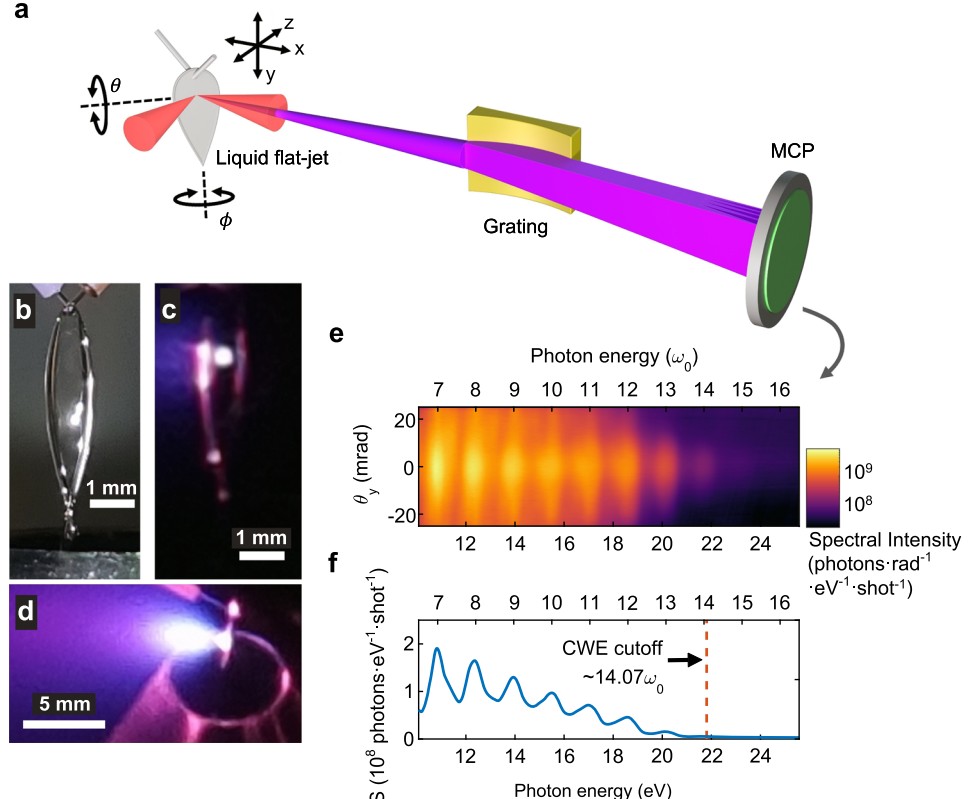

**Fig. 1 | High-harmonic generation from a liquid plasma mirror. a** Schematic drawing of the experimental setup. **b** Photo of a propylene glycol flat-jet target. **c** Photo of a propylene glycol flat-jet target exposed to laser pulses. The bright white spot is the fluorescence observed at the focus of the laser beam. **d** Photo of a propylene glycol flat-jet target and expanded plasma emerging along the direction normal to the liquid sheet. **e** Typical angle-resolved spectrum of CWE from a liquid plasma mirror. The plot is coloured in the logarithmic scale. **f** Spectrum of CWE obtained by integrating the divergence angle of the spectrum shown in **e**.

as shown in Fig. 2b. These results show that the CWE theory[3] can also be applied to HHG from a liquid plasma mirror.

The spectra recorded in the experiment for $\alpha = 0°$ (p-pol.), $\alpha = 45°$ and $\alpha = 90°$ (s-pol.) are shown in Fig. 2c–e, respectively (yellow cross-markers in Fig. 2a). This polarisation dependence is further evidence that the harmonic radiation observed was not generated from vapour molecules. If it had been generated from the vapour molecules, a considerable amount of the harmonic radiation would have been produced when the s-polarised laser field was used. In addition, the odd-order harmonics would have been stronger than even-order harmonics.

In contrast to HHG in gas molecules, CWE harmonics can also be generated using a circularly polarised laser field[59]. The CWE harmonics generated by a circularly polarised laser field (Fig. 2f) showed moderate intensity, comparable to that generated by a p-polarised laser field, showing the properties of the CWE harmonics. These observations safely exclude the possibility of gas-phase HHG in propylene glycol vapour. Therefore, the polarisation dependence observed in our experiments confirms that harmonic radiation was generated from the liquid plasma mirror.

## Dispersion dependence of CWE from liquid plasma mirrors

The CWE harmonic is produced in the plasma wake created by Brunel electrons. The CWE harmonic radiation is created at a specific time in every optical cycle[3,5]. The emission time depends on both the intensity and wavelength of the laser field. The emission time is advanced in a strong field and delayed in a weak field. Thus, the interval of the emission (or the separation of the attosecond pulses in time) is shorter in the rising part, whereas it is longer in the falling part of the laser field. These irregular emission times can be compensated when the laser

pulse is positively chirped[3–5,29,32]. Consequently, the spectral width of the harmonic radiation varies depending on the chirp condition of the driving laser field.

We recorded the CWE spectra for different amounts of GDD of the laser pulse (Fig. 3a). The dispersion was controlled using an acousto-optic programmable dispersive filter in the amplifier of the laser system. The temporal waveform and chirp of the laser pulse at each GDD value were measured by tunnelling ionisation with a perturbation for the time-domain observation of an electric field (TIPTOE) method[60] (see "Methods" section and Figs. S3–S4 in the Supplementary Information). The harmonic spectrum narrowed when the laser field was positively chirped, as shown in Fig. 3b. HHG from the liquid plasma mirror clearly showed the detailed structure of the spectra depending on the GDD values owing to the good stability of the light source. We performed 2D PIC simulations, and the results are shown in Fig. 3c, d. For a positive GDD value (400 fs²), the intensity of the spectrum was the brightest (Fig. 3c) and the spectrum was the narrowest (Fig. 3d). These results confirm that the CWE of a liquid plasma mirror can be controlled coherently.

## Intensity scaling of CWE from liquid plasma mirrors

We also studied the dependence of laser peak intensity on HHG on a liquid plasma mirror. The CWE harmonic spectra were recorded for different peak intensities, as shown in Fig. 4a. Each harmonic order shows a small but clear blueshift as the peak intensity increases. The observed photon number per laser shot of the harmonic radiation is shown in Fig. 4b as a function of the peak intensity. The observed photon number per laser shot was estimated from the sum of the spectrum from $6.5\omega_L$ to $14.5\omega_L$. The CWE photon number recorded in the experiment exhibited a linear dependence (Fig. 4b, red dashed line,

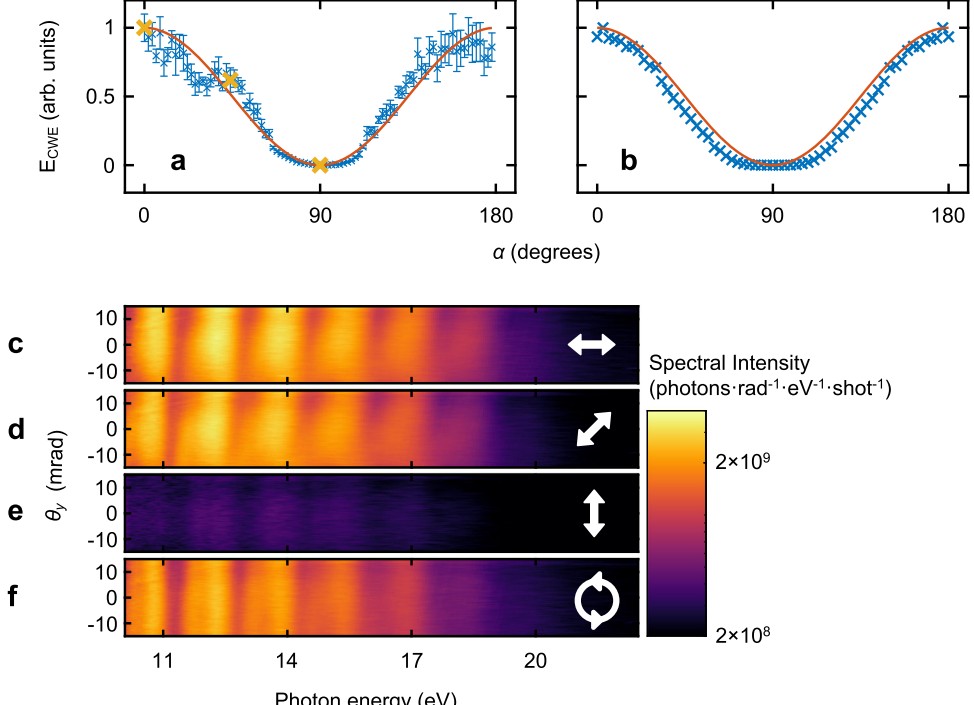

**Fig. 2 | Polarisation dependence of CWE generated from a liquid plasma mirror.** **a**, **b** Polarisation angle ($\alpha$) dependence of CWE spectral intensity obtained in experiments (**a**) and 2D PIC simulations (**b**). The p-polarisation corresponds to $\alpha = 0°$, and the s-polarisation corresponds to $\alpha = 90°$. The data points (blue crosses) and error bars show the mean values and standard deviations of 5 independent experiments, respectively. The solid red lines are cosine-squared functions for the comparison. **c**–**e** Angle-resolved CWE spectra from a liquid plasma mirror for three different polarisation angles, 0° (**c**), 45° (**d**) and 90° (**e**). **f** Angle-resolved CWE spectrum from a liquid plasma mirror for circularly polarised laser pulses.

$I_{CWE-EXP} \propto I_0^{1.1}$). The results of the PIC simulations, as shown in Fig. 4c, d, are consistent with the experimental results. The blueshift of the harmonic spectra was reproduced (Fig. 4c). The PIC calculations also show a linear dependence of the harmonic intensity on the laser intensity (Fig. 4d, red dashed line, $I_{CWE-PIC} \propto I_0^{1.2}$). The high-flux HHG from the liquid plasma mirror allowed us to observe the CWE dynamics with good statistics (see Figs. S5 and S6 in Supplementary Information).

### Relativistic high-harmonic generation from liquid plasma mirrors

As the laser intensity increases, high harmonic radiation is generated in the ROM regime. To generate ROM harmonics, the temporal contrast of the laser pulse should be sufficiently high. Otherwise, the pre-pulses can create a pre-plasma too early. In this case, the plasma scale length becomes too long or the plasma surface can be spoilt before the main pulse arrives (see Fig. S7 in Supplementary Information). Therefore, we used a 150-TW-laser system[58] with high temporal contrast to demonstrate HHG from the liquid plasma mirror in the ROM regime. The 150-TW-laser beams were delivered to the liquid target via double plasma mirrors. The temporal contrast of the laser pulse was better than $10^{11}$ before 3 ps of the main pulse. A small part of the laser beam was used to create plasma on the liquid flat-jet (see Fig. S8 for the beam path diagram in Supplementary Information). We optimised ROM HHG by controlling the intensity and timing of the pre-pulse. We observed the best ROM HHG when the pre-pulse arrived 2 ps earlier than the main pulse. The peak intensity and fluence of the pre-pulse were ~$10^{16}$ W/cm² and 800 J/cm². Under these conditions, the plasma scale length estimated by the PIC simulation was around $0.1\lambda$. The HHG spectrum was recorded from the liquid plasma mirror in the specular direction (Fig. 5). The spectrum was observed up to $33\omega_0$ (51.1 eV), which is significantly higher than the CWE cutoff of $14.1\omega_0$ (21.8 eV). Each harmonic peak was clearly observed. This observation indicates that the

liquid target is reasonably flat on the scale of sub-wavelength scale of harmonics[61]. The peak intensity of the laser pulse was estimated to be $1.7 \times 10^{20}$ W/cm², which was considerably higher than the relativistic intensity. Thus, we conclude that the observed spectrum was generated in the ROM regime. These experimental results confirm that the liquid plasma mirror can be used to produce stable and high-flux harmonic radiation in both CWE and ROM regimes.

## Discussion

Ultrafast light-matter interactions have been extensively investigated over the last few decades. Attosecond pulses generated through the recollision dynamics in atoms have been a workhorse in this field, providing high repetition rate light sources for time-resolved studies. In addition, ultrafast phenomena in atoms, molecules, liquids and solids have been widely investigated in high-harmonic spectroscopy[62]. However, the conversion efficiency of recollision HHG[63] is known to be ~$10^{-8}$–$10^{-5}$. In addition, the intensity of the driving laser field is limited to ~$10^{14}$–$10^{15}$ W/cm² because of the depletion of the ground state. With these experimental parameters, attosecond pulses can be generated with energies of ~nJ or ~μJ. Consequently, their applications have been mainly limited to photoelectron spectroscopy in the linear regime (i.e. photoelectron spectroscopy), except for a few pioneering works[64,65].

We demonstrated the continuous generation of harmonic radiation with a liquid plasma mirror. The conversion efficiency of the 7–14th CWE harmonics shown in Figs. 3 and 4 was estimated to be $1 \times 10^{-4}$ (see Figs. S9–S14 in the Supplementary Information and the "Method" section for the calibration process), which shows a reasonable agreement with the conversion efficiency obtained with solid targets[3,4]. We also performed PIC calculations with different electron number densities. The conversion efficiency is not strongly dependent on the electron number density in both CWE and ROM regimes (see Fig. S14 in the Supplementary Information). These results show that

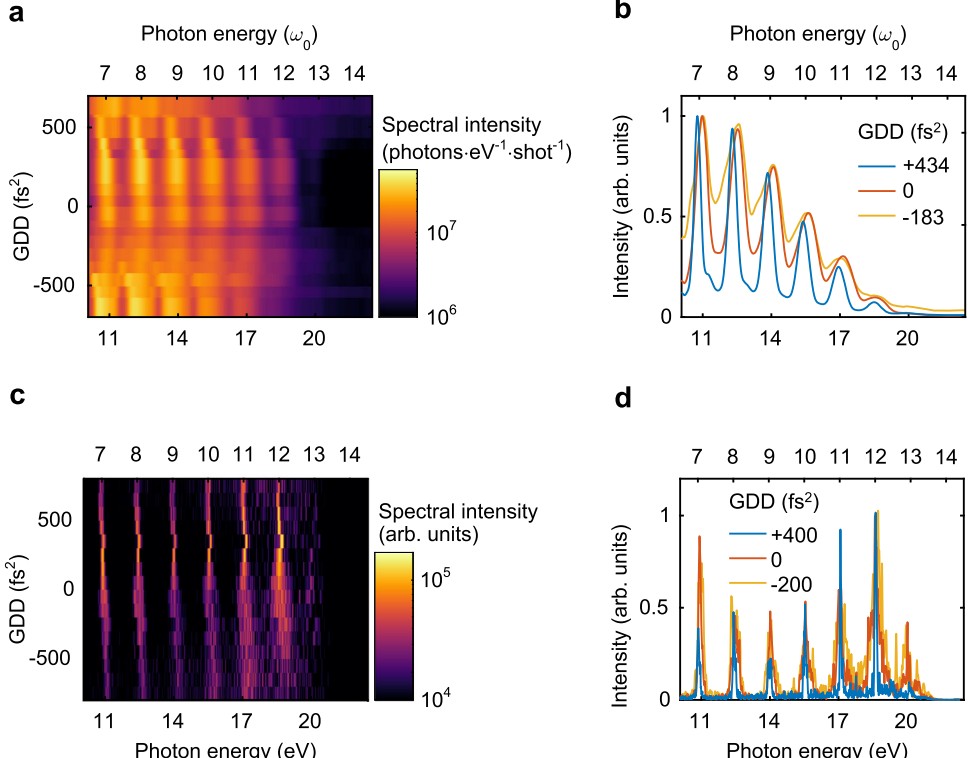

**Fig. 3 | Dispersion dependence of CWE from liquid plasma. a** Angle-integrated CWE spectra obtained in the experiment for the 21 different GDDs of a driving laser field. The spectra are the mean values of five independent experiments. **b** Comparison of three spectra shown in **a**. Each spectrum was normalised to a maximum intensity between $6.5\omega_0$ and $14.5\omega_0$ for better comparison of the spectral width. **c** Angle-integrated CWE spectra obtained in PIC simulations in 2D. **d** Comparison of three spectra shown in **c**. Each spectrum was normalised to a maximum intensity between $6.5\omega_0$ and $14.5\omega_0$ for better comparison of the spectral width.

the high conversion efficiency of solid-plasma sources is translated to the lower-density liquid-plasma.

We obtained CWE harmonic radiations continuously at a repetition rate of 1 kHz by using a liquid flat-jet. In this experiment, the liquid plasma mirror was operated continuously over 5 h, which corresponds to 18 million consecutive laser shots. We also confirmed that it can be operated for a longer time by circulating the liquid or using a larger volume of the liquid reservoir (see Fig. S1 in the Supplementary Information). Such high repetition rate operation and data collection would enable precision measurements and provide large sets of data. Thanks to the continuous operation of the liquid flat-sheet plasma mirror at high repetition rates, we were able to observe the structure of the CWE harmonics for various laser parameters, including polarisation, ellipticity, dispersion and energy, all in a single day. This achievement would have been challenging if a target supports a limited number of laser shots. The HHG spectra from the liquid plasma mirror is the outcome of the interaction of the ultra-intense laser field with the plasma. Thus, the results of this study pave the way for further investigation of ultrafast light-matter interaction in a plasma state using conventional high-harmonic spectroscopy techniques with stable, high-flux harmonic pulses[66]. Also, a large data set would enable applications of machine learning and artificial intelligence methods for multi-variable optimisation of attosecond pulse generation from a plasma mirror[67].

We demonstrated HHG from the liquid plasma mirror in the ROM regime using a 150-TW-laser with a pulse energy of 5 J. According to the existing research[17,21,68], the conversion efficiency of HHG from a plasma mirror varies depending on the harmonic order $q$ as $q^{-p}$ with the exponent $p = 4/3 \sim 8/3$. Thus, attosecond pulses can be obtained with energies of ~mJ or even higher. These estimations indicate that the liquid plasma mirror is a promising light source for the generation of

intense, high-repetition-rate attosecond pulses. Consequently, the results of this study will significantly extend the applicable areas of HHG from a plasma mirror, including time-resolved pump-probe studies and attosecond streaking experiments in both the linear and nonlinear regimes.

It is also worth mentioning that the liquid plasma mirror does not solve all the issues of high-power, high-repetition laser experiments. The temporal contrast of the high-power laser pulse is critical in these experiments (see Fig. S7 in the Supplementary Information). Another liquid plasma mirror that can be operated at a high repetition rate would be required to improve the temporal contrast of high-repetition-rate laser pulses. Also, the limited repetition rate of high-power laser systems should be improved. Several optical parametric chirped pulse amplification systems already show a good performance. Some of these laser systems can produce laser pulses with a pulse duration below 10 fs and an energy of several tens of milli-Joules at a repetition rate of 1 kHz[69,70]. Thus, these laser systems can provide a laser pulse with a high peak intensity on the target that can reach up to ~$10^{20}$ W/cm$^2$. Therefore, intense, high-repetition-rate EUV and X-ray pulses can be obtained through HHG from a liquid plasma mirror.

## Methods
### Liquid flat-jet target
We degassed propylene glycol before the experiments because any dissolved gas in the liquid hinders the stable operation of the liquid flat-jet in a vacuum. The liquid was degassed in a bell-jar vacuum chamber until the pressure in the bell-jar chamber reached around 300 mtorr and no bubble was observed. The degassed liquid was stored in a stainless beaker with a volume of 5 L (see Figs. S1–S2 for a complete connection diagram in the Supplementary Information). In order to pump propylene glycol into the target chamber, we used a dual-piston,

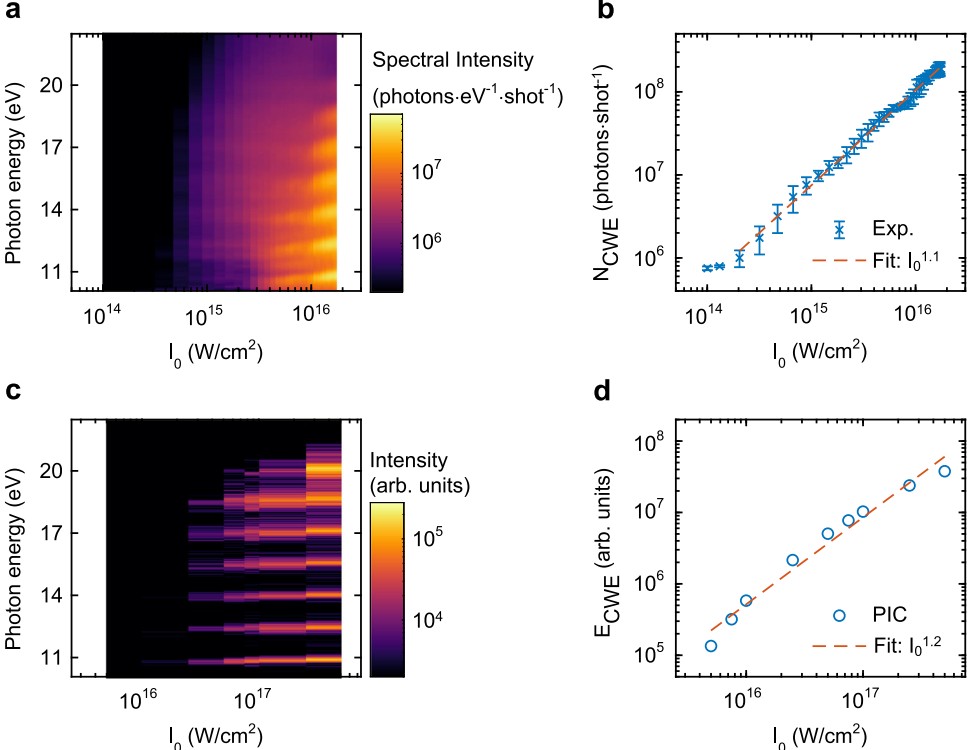

**Fig. 4 | Intensity scaling of CWE from a liquid plasma mirror. a, b** Angle-integrated CWE spectra (**a**) and the sum of the observed spectral intensity of the CWE spectra (**b**) between $6.5\omega_0$ and $14.5\omega_0$ obtained in experiments for different peak intensities of the driving laser. The data points are the mean values of five independent measurements. The error bars show the standard deviation. **c**, **d** Angle-integrated CWE spectra (**c**) and the sum of the intensity of the CWE spectra (**d**) between $6.5\omega_0$ and $14.5\omega_0$ obtained in 2D PIC simulations obtained at the different peak intensities of the laser pulse. Note that the intensity ranges of experimental results (**a**, **b**) and PIC simulations (**c**, **d**) are different because the experimental results could not be observed at high intensities due to the low temporal contrast of the laser pulse.

high-pressure metering pump (Vindum Engineering, Inc., VP-12k). Each syringe of the pump has a stroke volume of 10 ml and the exchange between the pistons is smooth, which is capable of pulse-free continuous operation for a long experimental time. The pump was operated at a constant flow rate mode. The flow rate was set to 14 ml/min and the pressure was kept at ~4000 psi. In order to reduce fluctuations in the liquid pressure within a few seconds down below 0.1%, we used two types of pulsation dampeners. One was the in-line dampener connected in serial to the tube, and the other was three vertical tubes with a blanked end connected in parallel to the tube. In this way, the fluctuation in pressure was reduced. The stability of the liquid flat-jet target was improved significantly.

The liquid flat-jet target was created by colliding jets from two capillaries with an inner diameter of 150 μm. Each capillary was connected to a metering valve to match the flow rates through the two capillaries. The capillary tubes were mounted on aluminium parts with a fixed colliding angle. We tried various colliding angles, 60°, 90°, 120°, 135° and 150°. We chose 90° as the colliding angle because the stability and the size of the liquid flat-jet were optimal. The liquid flat-jet target was mounted on a combination of motorised stages, 3D translation stages (Zaber, LSM025A-V1T4), a rotary stage (Zaber, X-RSM40B-SV1) and a goniometer (Zaber, X-RSM60-SV1). With this combination of stages, the position and angle of the liquid flat-jet target were controlled accurately. The liquid flat-jet fell and collected at the bottom of a liquid catcher (see Figs. S1–S2 for liquid connection and the catcher design in the Supplementary Information). The catcher was pumped by a turbo molecular pump to keep the low pressure level. A silicone tube was connected to the bottom of the catcher and the liquid was pumped out into the air by a peristaltic pump. The used liquid was collected in another liquid reservoir in the air.

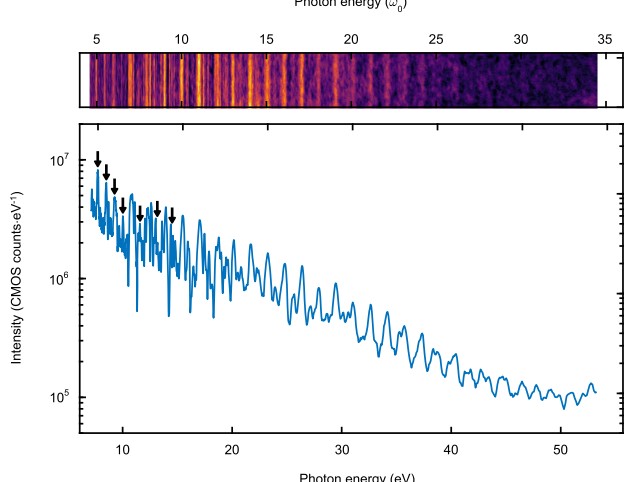

**Fig. 5 | High-harmonic spectrum generated in the ROM regime using a liquid plasma mirror.** The upper panel shows the angle-resolved spectrum, and the lower panel shows the angle-integrated HHG spectrum generated through ROM from a liquid plasma mirror. The black arrows denote the second-order diffraction of the harmonics.

## Experimental setup for 1-kHz coherent wake emission experiment

In the 1-kHz CWE experiments, we used a Ti:sapphire laser system (Femtopower X CEP, 10 mJ, 30 fs, 800 nm, 1 kHz). The temporal contrast of the output laser pulse was $10^6$ at 4 ps before the main pulse, which was measure by tunnelling ionisation with a

perturbation for the time-domain observation of an electric field (TIPTOE)[60]. We placed two half-wave plates in the beam path (see Figs. S15–S16 for the beam path diagram in the Supplementary Information). One was placed before a pair of thin film polarizers to control the energy of the pulses and the other was placed after the polarizers to control the polarisation angle of the linearly polarised pulses. Each half-wave plate was mounted on a motorised rotary stage (Zaber, X-RSW-SV1) to control the angle accurately. The beam size was expanded after a pair of thin film polarizers to 20 mm in diameter. A quarter-wave plate was placed after the second half-wave plate and the angle of the quarter-wave plate was fixed to make a circularly polarised laser field. The quarter-wave plate was used to obtain the results shown in Fig. 2f and the quarter-wave plate was not in the beam path otherwise. The beam was sent to the target chamber. The beam was focused by a 90° off-axis parabolic mirror (OAP) with a focal length of 5 cm (see Fig. S17 for focal spot images in the Supplementary Information). In order to control the GDD of the laser pulses, we changed the setting value for GDD of the acousto-optic programmable dispersive filter in the laser system. The temporal profiles of the laser pulses for 21 different values of the GDD were measured by using TIPTOE (see Figs. S3–4 in the Supplementary Information). The beam was intercepted by placing a mirror before the OAP and the intercepted beam was aligned to the TIPTOE setup, which was placed next to the target chamber. In order to send the beam to the TIPTOE setup, a side wall of the target chamber was removed. We measured the temporal profiles 5 times for each GDD. The intercepting mirror was removed after the TIPTOE measurements. The CWE beam was aligned to a flat-field EUV spectrometer consisting of a grating, a microchannel plate (MCP) and a scientific complementary metal–oxide–semiconductor (sCMOS) camera. We used a flat-field, concave, aberration-corrected EUV grating with a groove density of 120 grooves/mm (Shimadzu, 30-007). The distance between the focal spot and the grating was 450 mm. The MCP has a phosphor screen as a back plate. Images of the phosphor screen were taken by the high dynamic range sCMOS camera (Excelitas PCO GmbH, pco.edge 5.5, 16 bit, 2560 × 2160), which was synchronised to the arrival of the laser pulse. The exposure time was adjusted between 60 ms to 200 ms depending on the intensity recorded by the camera.

**Experimental setup for relativistic oscillating mirror experiment**
In the ROM experiment, we used a 150-TW Ti:sapphire laser system. Detailed descriptions about the laser system can be found in the reference[58]. We briefly describe the key features of the system. The temporal profile was measured by spectral phase interferometry for direct electric-field reconstruction (SPIDER) and the pulse duration was 29 fs in full width at half maximum (FWHM). We employed double plasma mirrors to achieve a high temporal contrast better than $10^{11}$ before 3 ps of the main pulse. The wavefront aberration of the beam after the double plasma mirrors was corrected by a deformable mirror and a wavefront sensor (AKA optics) to make the best focal spot at the target position. The beam after the deformable mirror was sent to the 30° OAP and the beam was focused on the liquid flat-jet target (see Fig. S9 in the Supplementary Information). The focal spot was imaged by a set of lenses and a charge-coupled device (CCD) camera (see Fig. S17 for focal spot images in the Supplementary Information). The imaging system was mounted on 3D translation stages. The imaging system was withdrawn on laser shots with full energy. The system was covered by a motorised stage to block any debris or splattering liquid droplets during the experiment. The reflected beam was aligned to another flat-field EUV spectrometer. The arrangement of the spectrometer is the same as the one used in 1-kHz CWE experiments but the groove density of the grating was 300 grooves/mm (Shimadzu, 30-006). Each angle-resolved spectrum was recorded for a single laser shot.

**Calibration of photon counts**
To calibrate a photon number observed in experiments, we recorded CWE spectra using a spectrometer equipped with an X-ray CCD camera (Teledyne Princeton Instruments, PIXIS XO 2048B, see Fig. S9 in the Supplementary Information). Then, we recorded the CWE spectrum under the same condition using a spectrometer equipped with an MCP detector, which was used in the main experiments. The photon numbers recorded by the CCD were compared to the counts recorded by the MCP spectrometer.

The X-ray CCD was attached to the wall of the chamber. An aluminium blade was placed to block the specular reflection from the grating. The distance from the focal spot to the grating was 830 mm. To block the laser beam, we placed two aluminium filters in the beam path. The first filter has a thickness of 500 nm, and the other filter has a thickness of 200 nm. The spectrum was recorded in the low-noise output mode with the pixel-to-pixel readout rate of 100 kHz. The gain of the CCD was set to the maximum value. With these settings of the CCD, the conversion gain from an analogue-to-digital unit (ADU) to the number of electrons, $e_{ADU}$, was 0.88 e⁻/ADU and the nonlinearity was <2%. The exposure time was 42 s, which was the time needed to read all the pixels of the CCD. Three harmonics were recorded, where the highest order of harmonic was cut by an aluminium blade (see Fig. S9 in the Supplementary Information). The counts recorded in the shadow of the blade were used to estimate the noise level of the CCD. Considering the transmittance of the aluminium filters, we determined that the lowest order observed in the X-ray CCD was 11th harmonic (see Fig. S11 in the Supplementary Information). Because the 12th harmonic was the brightest in the recorded image, we chose the 12th harmonic to calibrate the number of photons. The average ADU count at the peak of the 12th harmonic, $C_X$, was 3000. The quantum efficiency of the CCD at 18.6 eV (12th harmonic), $q_e$, was ~33%. The number of photons incident to the pixel of the CCD at the peak of the 12th harmonics was estimated to be 7928. The number of photons per unit area was obtained by dividing the number of photons per pixel by the area of the pixel, $p_X^2$, where $p_X = 13.5\,\mu m$ is the size of a pixel of the CCD. To estimate the photon number before transmitting through the aluminium filters, the incident number of photons per unit area was divided by the transmittance (0.48%) of the aluminium filter (Al 660 nm + Al₂O₃ 40 nm), $T_{Al}$. We referred to the database for the transmittance of the aluminium filter at 18.6 eV[71]. We divided the number obtained up to this point by the number of accumulated laser shots, $N_X$, to obtain the number of photons per unit area per shot, $n_X$, and its expression can be written as $n_X = (C_X e_{ADU})/(N_X q_e T_{Al} p_X^2)$. We obtained the count per unit area per shot observed in the MCP spectrometer $n_M = C_M/(N_M p_M^2)$, where $C_M$ is the count at the peak of the 12th harmonic recorded by using the MCP spectrometer, $N_M$ is the number of laser shots accumulated to obtain an MCP image, and $p_M$ is the scale of the image per pixel. We divided $n_X$ by $n_M$ and multiplied it to $(d_X/d_M)^2$, where the factor $(d_X/d_M)^2$ corrects the difference in irradiance due to the different distances between the source and the grating in each setup. Thus, the calibration factor $c_f$ that converts from MCP counts to the photon number observed by the MCP spectrometer can be written as Eq. (1)

$$c_f = \frac{n_X}{n_M}\left(\frac{d_X}{d_M}\right)^2 = \frac{C_X e_{ADU} N_M}{C_M N_X q_e T_{Al}}\left(\frac{p_M}{p_X}\right)^2\left(\frac{d_X}{d_M}\right)^2 \tag{1}$$

The calibration factor was estimated to be $c_f = 7.7$ (photons/CMOS counts). The calibration factor $c_f$ is obtained using many parameters ($C_X$, $C_M$, $q_e$ and etc.) that have errors. For example, the uncertainty of the transmittance estimation on the oxidation layer thickness of the filters is 40%. Therefore, the photon count presented in this work should be accepted as a representation of an order of magnitude, not an exact value.

## Estimation of conversion efficiency

To estimate the conversion efficiency (see Fig. S12 in the Supplementary Information), we first obtained the energy of the observed harmonics between the $7^{th}$ harmonic and 14th harmonic, $E_{observed} = \iint \omega S(\omega, \theta_y) \sin\theta_y / g_e(\omega) d\omega d\theta_y$, where $\omega$ is photon energy, $S$ is spectral intensity, $\theta_y$ is cone angle, $g_e$ is diffraction efficiency of the grating, and the integration over $\theta_y$ was taken for the recorded range, $\pm 25$ mrad. To estimate the full energy of the CWE beam, $E_{CWE}$, we calculated $E_{observed} A_{CWE} / A_{grating}$, where $A_{CWE}$ is the area of the CWE beam at the grating and $A_{grating}$ is the open aperture area of the grating. We assumed the divergence of the CWE beam to be $\pm 100$ mrad in cone angle (see Fig. S13 in the Supplementary Information). We estimated the ratio $A_{CWE}/A_{grating}$ to be 81. The conversion efficiency was obtained by dividing the estimated full energy of the CWE beam by the input energy of the laser pulse. Considering the uncertainty of the calibration factor, grating efficiency and the estimation of the beam divergence, the conversion efficiency presented in this work should be accepted as a representation of an order of magnitude, not an exact value.

## Particle-in-cell simulations

We performed two-dimensional (2D Cartesian) particle-in-cell (PIC) simulations by using SMILEI code[72]. The size of the spatial domain was $50\lambda \times 50\lambda$ ($12801 \times 12801$ grid). The time step size was $T_L/370$. We used the Silver-Müller absorbing boundary condition for electromagnetic fields. The plasma consists of a homogeneous region and an inhomogeneous region. The homogeneous region was $45\lambda \times 10\lambda$ ($W \times T$) rectangle inclined to the $x$-axis by 45°. The centre of the front surface was placed at ($25\lambda$, $25\lambda$). The homogeneous part has a uniform plasma density of $198n_c$. The inhomogeneous region has an exponentially decaying profile with the scale length ($L = 0.006\lambda$) optimised for CWE, where the decaying profile starts from the front edge of the homogeneous region and ends $10L$ away from the front edge. We used electron-proton plasma. The number of macro particles per cell for the electron was 36, and that for the proton was 9. A 30-fs (FWHM), 800-nm pulse was focused along the $x$-axis. In this way, the laser beam was incident to the plasma with an incidence angle of 45°. The $1/e^2$ radius of the beam waist was $3.2\lambda$. The peak intensity of the laser field at the focal spot was $5 \times 10^{16}$ W/cm$^2$ for linear polarisation and circular polarisation simulations. In GDD simulations, the peak intensity was adjusted accordingly to the pulse duration at each GDD, where the peak intensity was $1 \times 10^{17}$ W/cm$^2$ for the chirp-free pulse. In peak intensity scaling simulations, the peak intensities were from $5 \times 10^{15}$ W/cm$^2$ to $5 \times 10^{17}$ W/cm$^2$. Probes for the electromagnetic field were placed in a specular direction. The probes were arranged horizontally from ($10\lambda$, $43.5\lambda$) to ($40\lambda$, $43.5\lambda$). The spatio-temporal profiles of the reflected electromagnetic fields were recorded by the line of probes. A temporal window was applied to the profiles to exclude noisy radiation from plasma after interaction with the laser field. Then, the angle-resolved spectrum was obtained by the Fourier transformation in 2D (one in space and the other in time).

## Data availability

The data that support the plots within this paper and other findings of this study are available from the corresponding author upon request.

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

## Acknowledgements

This work was supported by the Institute for Basic Science grant (IBS-R012-D1). This work was supported by the National Research Foundation of Korea (NRF) grant funded by the Korea government (MIST) (No. 2022R1A2C3006025). Computational works for this research were performed on the IBS Supercomputer, Aleph in the IBS Research Solution Center.

## Author contributions

Y.H.K. and K.T.K. conceived and designed the experiment. Y.H.K., S.C.P., Y.K., K.Y., W.C., T.K., H.Y., J.H.S. and S.K.L. performed the experiment.

H.K. performed the PIC calculations. Y.H.K., H.K. and K.T.K. analysed the experimental and PIC calculation results. T.T.L. participated in designing the flat liquid jet and the vacuum chambers. Y.H.K., C.H.N. and K.T.K wrote the manuscript.

## Competing interests

The authors declare no competing interests.
