## [Peer Review File · Nature Communications]

High-harmonic Generation from a Flat Liquid-sheet Plasma MirrorEditorial Note: This manuscript has been previously reviewed at another journal that is not operating a transparent peer review scheme. This document only contains reviewer comments and rebuttal letters for versions considered at *Nature Communications*.

REVIEWER COMMENTS

Reviewer #1 (Remarks to the Author):

The revised manuscript includes new measurements of the photon flux and stability of the CWE harmonic source. Based on the high conversion efficiency and reasonable stability, I feel that it is reasonably likely that such a source could enable novel measurements that are currently out of reach for both neutral gas and solid plasma harmonic sources. Although I am still somewhat skeptical about whether the degree of novelty is sufficient to merit publication in a high-impact journal, I remain impressed by the technical quality of the work and I am willing to support publication in *Nature Communications*.

Reviewer #4 (Remarks to the Author):

The manuscript "High-harmonic Generation from a Liquid Plasma Mirror" by Yang Hwan Kim et al. reports on a series of experiments of surface-high-harmonic generation on plasma mirrors, created on the surface of liquid-sheet targets.

The majority of the paper presents fairly complete experiments that demonstrate the majority of the well known properties and behaviors of CWE harmonics. This shows that on the liquid sheet target, CWE-HHG with its requirement for very steep density gradients, behaves "just the same" as we know it from optically polished solid targets. So while this does not yield insight into "new physics", it convincingly benchmarks the usability of liquid sheet targets for surface-HHG. This usability was far from obvious before: the results shown in the only preceding publication on surface-HHG on a liquid target, ref. 56, did clearly show far-sub-optimal performance and did not spark great motivation to try the same in your own lab. The results reported here are very different, very motivating for everybody working on plasma mirrors and the interaction of overcritical plasmas with lasers, and to me represent an important milestone.

The section on ROM-HHG in the relativistic regime is unfortunately very sparse. We are shown on a single spectrum and are told almost nothing about the precise interaction conditions. So it remains largely unclear how "optimal" the achieved conditions for ROM were.

On p.11, line 217, we find the only mention of a pre-pulse used in the ROM experiments for the plasma density gradient control, and no other details are given in the methods section. So, at what delay was this prepulse arriving on the target, and with what fluence? Do the authors have an estimate of the generated plasma-scale length? Do they think they were close to optimal conditions for ROM or just "good enough to get a signal"? How is the signal without a prepulse?

The main merit of the work is that it presents a clear technological breakthrough that opens many new perspectives and answers many technical questions. So for the field of ultra-intense laser-plasma interaction, and the small community working on plasma mirrors in particular, this is a major result, which I think deserves publication in a journal like Nature Communications.

But there are also some serious technical shortcomings left that need to be addressed before I can recommend publication:

Problematic comparison of experimental and PIC-simulation results in Fig. 4:

When discussing Fig.4 with the intensity dependence of the CWE harmonics, the paper states "The results of the PIC simulations, as shown in Figs. 4c and 4d, are consistent with the experimental results." This statement seems problematic to me, because the PIC simulations were made for higher intensities than the experimental ones. The simulated intensity range in Fig. 4 overlaps only for less than an order of magnitude with the experimental intensity range. So there are actually only the first 3 simulation-data-points in the experimental range, and they have a steeper intensity dependence than the linear fit. So to me the agreement is not very good.

Also to me the PIC-simulation data points in Fig.4d suggest a sub-linear intensity dependence over the simulated intensity range.

This comparison needs to be explained better, the different intensity range must be justified.

The way it is presented currently is misleading because different intensity ranges are presented on top of each other as if they were the same. The different ranges must appear clearly in the graphical presentation, or at the very least be mentioned in a warning in the caption!

I am not convinced by the photon count calibration that has been added to the paper.

- The first crucial issue is the lack of any uncertainty estimate. This must be added for both the calibration factor $c_f = 7.7$ (photons/CMOS counts) as well as for the conversion efficiency. This must account for the uncertainty of the various geometrical corrections and the precision of other assumptions that went into this calibration (thickness of oxidation layer on the filters, MCP spectral response, etc...).

- I also disagree with some of the assumptions made for the estimation of the conversion efficiency on p.20:

- The total energy of the CWE beam is found by first summing up all the photons' energies that make it onto the MCP detector. Then this "observed energy" is simply scaled up by the ratio of the area of the beam cross section at the grating over the much smaller open aperture area of the grating. But that seems to implicitly assume that the average fluence outside of the observed part of the beam - notably at the edge of the 100mrad cone - is the same as in the observed part. Such a roughly homogeneous distribution, or roughly top-hat beam, seems rather unrealistic and to me seems to lead to a clear overestimation of the conversion efficiency, doesn't it?

Shouldn't one rather consider that the grating effectively only focuses a thin 1D-slice out the polar beam distribution, and therefore one would have to integrate over $\theta d\theta$, as was done in [Kahaly et al. PRL 110, 175001 (2013)] ?

- The total CWE beam divergence was assumed to be ± 6100 mrad, based purely on a 2D PIC simulation. But a spectrum like that in Fig. 1e should allow an experimental estimation of the divergence, even though the detected angular part of the beam is narrow? Or can the experiments at least corroborate this very large ± 100 mrad value? The larger this assumed value is, the larger of course the extracted conversion efficiency becomes - so again I am wary of an overestimation of the conversion efficiency here. At the very least, this divergence estimate should contribute to an important uncertainty on the experimental conversion efficiency!

- The authors vaguely suggest on page 4 that only with liquid targets, kHz-rep-rate operation of plasma mirrors is possible. That is not true, as kHz-operation in the CWE regime has been demonstrated as early as 2011, and this work is cited as refs. 26, 29, 32, 34, 38. Most recently, the relativistic regime and ROM-HHG at kHz-rep-rate has also been demonstrated by the same group: [Haessler et al. Ultrafast Science 2022, 9893418 (2022)]. In that paper, detailed experimental parameter scans of the plasma density

gradient scale length where reported. So the authors should explicitly state that kHz operation in both sub-relativistic and relativistic regimes has been demonstrated, and perfectly meaningful parameter scans are made. Once this state-of-the-art is clearly laid out, there is still a very clear case to be made why liquid sheet targets and their continuous operation for hours are far superior - be it for very extensive acquisitions like the mentioned streaking measurements, or simply for an operation of a secondary XUV source for actual applications.

- Connected to the preceding point: in the conclusions on p.13, line 256, the authors state that the high rep-rate and long-term operation of their liquid-sheet target "allows us to analyse the detailed structure of the CWE harmonics under various conditions, such as polarisation, ellipticity, dispersion, and energy, thanks to the good stability of the light source." Their results are certainly nice, but all these studies have been made before with solid targets. So if the authors want to make such claims, they need to argue precisely what has been possible up to now, including the demonstrated 1-kHz operation with solid targets, and what will become possible only thanks to liquid-sheet targets.

- The authors vaguely state (on p.4 line 71) that rotating disks and tape targets "suffer from mechanical instability at high-repetition-rate operations". This is a "null statement", because it remains open what "instability" is supposed to mean. What stability requirement for which application is not reached with solid targets? The authors cite ref. 34 where active stabilization (albeit for a ridiculously short amount of time) has been shown to enable interferometric stability, i.e. $\sim 100\text{nm}$ position stability of a rotating disk. It is not clear that a liquid sheet target could do any better. So I don't think the authors can so easily claim, without hard evidence, that a liquid sheet target would solve any stability issues.

- A similar vague reference to "stability" is made when discussing Fig. 3b (p.9, line 186), when it is stated that it "showed the detailed structure of the spectra depending on the GDD values owing to the good stability of the light source". Again: what precisely is meant by that? What kind of instability would blur the spectral structures?

Minor remarks:

- "Liquid plasma mirror" in the title seems a bit too general, given that there has been one experiment with a liquid jet before [Heissler et al. New Journal of Physics 16, 113045 (2014)]. I suggest specifying "flat liquid-sheet" or so.

- The formulation "Its usage is limited to applications such as time-resolved studies and attosecond streaking experiments..." is misleading, because this passive form means that "only such applications are possible". The contrary is true: no such experiments could be done with plasma mirror harmonics so far. So I suppose the authors mean that its usage limits, or even prevents such applications. So: the active form "limits"/"prevents" should be used rather than the passive "is limited to".

- On p.10, line 197, the sentence "The spectral intensity was estimated from the sum of the spectrum from $6.5\omega_L$ to $14.5\omega_L$." is not clear to me. If one "sums a spectrum" along its spectral dimension, one no longer has "spectral" intensity, right? I suppose this refers to the ordinate axis of Fig. 4b, which shows photon number/shot, i.e. not at a "spectral intensity" but something corresponding to a pulse energy. The description of panel 4b as "the sum of the peak intensity of the CWE spectra (b) between $6.5\omega_L$ to $14.5\omega_L$ " given in the caption of Fig.4 is equally incomprehensible to me, and I am left guessing what the connection to the photons/shot may be. This must be cleared up.

- On p.11, line 225, "the ROM model" is mentioned, without a reference. The authors should not suggest that there is a canonical "ROM model", because that is untrue. There has been a period of time when the theory of ref.17 was considered to provide a "universal spectral shape" of $q^{-8/3}$ (used to fit the exp. ROM spectrum in Fig.5), and a fundamental understanding. Today, our understanding has much advanced, problematic assumption in the Baeva-model are recognized, and we know about a greater complexity of the involved relativistic plasma dynamics. Much less steep spectral shapes $q^{-4/3}$ are predicted, e.g. in refs. 21 and 67.

- On p.13, line 265, the authors write "We also showed that the harmonic intensity increases linearly as the peak intensity of the driving laser pulse increases." This must refer to Fig.4 with CWE results, but

starts the paragraph about the ROM-results, which is confusing. Also, this mention suggests that this linear scaling (i.e. constant conversion efficiency) would be a new results, which it is not: see Fig. 1 in [Quére et al. Physical Review Letters 96, 125004 (2006)].

- At the end of the conclusions, on page 14, ROM-HHG with a kHz-driver is mainly presented as an outlook, and new kHz-lasers with tens of mJ pulse energies are mentioned. But the state of the art includes ROM at 1kHz rep-rate with few-mJ drivign pulses, as demonstrated in [Haessler et al. Ultrafast Science 2022, 9893418 (2022)].

- In the description of the PIC simulations on p.21, the authors mention the plasma density scale length of 0.006λ . This is an extremely short value - how was this chosen? CWE is typically sai to be optimized for a $\lambda/100$ to $\lambda/50$ scale length (see e.g. ref.3)- so 2 or 3 times longer.

It is clearly unrealistically short for an experiment. Even if the peak intensity was only $1e16$ W/cm², the contrast ratio was also only $1e6$. Have the authors done simulations with other longer scale lengths? How robust was the agreement of the linear-polarization scan (Fig.2) with the experiment for different simulated conditions?

Why are the harmonics peaks in the simulations in Fig.3 so much more narrow than in the experiments? Maybe because the gradient scale length is longer in the experiments?

- In the caption of Fig.5, "5 independent experiments" are mentioned. What exactly means 'independent'? 5 successsive acquisitions? Or from 5 different days with their individual alignment etc.? How does a scan like Fig; 3a look without such averaging?

Response to referees

- * The reviewer's comments are shown in blue.
- * Our answers are shown in black.
- * The modifications made in the revised manuscript are shown in red.

Reviewer #1 (Remarks to the Author):

The revised manuscript includes new measurements of the photon flux and stability of the CWE harmonic source. Based on the high conversion efficiency and reasonable stability, I feel that it is reasonably likely that such a source could enable novel measurements that are currently out of reach for both neutral gas and solid plasma harmonic sources. Although I am still somewhat skeptical about whether the degree of novelty is sufficient to merit publication in a high-impact journal, I remain impressed by the technical quality of the work and I am willing to support publication in Nature Communications.

We sincerely appreciate the referee's positive evaluation of our manuscript and recommendation for its publication.

Reviewer #4 (Remarks to the Author):

The manuscript "High-harmonic Generation from a Liquid Plasma Mirror" by Yang Hwan Kim et al. reports on a series of experiments of surface-high-harmonic generation on plasma mirrors, created on the surface of liquid-sheet targets.

The majority of the paper presents fairly complete experiments that demonstrate the majority of the well known properties and behaviors of CWE harmonics. This shows that on the liquid sheet target, CWE-HHG with its requirement for very steep density gradients, behaves "just the same" as we know it from optically polished solid targets. So while this does not yield insight into "new physics", it convincingly benchmarks the usability of liquid sheet targets for surface-HHG. This usability was far from obvious before: the results shown in the only preceding publication on surface-HHG on a liquid target, ref. 56, did clearly show far-sub-optimal performance and did not spark great motivation to try the same in your own lab. The results reported here are very different, very motivating for everybody working on plasma mirrors and the interaction of overcritical plasmas with lasers, and to me represent an important milestone.

We are grateful for the referee's thoughtful evaluation of our manuscript. We have carefully considered the referee's insightful suggestions and comments. We have made corresponding modifications to our manuscript, as detailed below.

The section on ROM-HHG in the relativistic regime is unfortunately very sparse. We are shown on a single spectrum and are told almost nothing about the precise interaction conditions. So it remains largely unclear how "optimal" the achieved conditions for ROM were.

On p.11, line 217, we find the only mention of a pre-pulse used in the ROM experiments for the plasma density gradient control, and no other details are given in the methods section. So, at what delay was this prepulse arriving on the target, and with what fluence? Do the authors have an estimate of the generated plasma-scale length? Do they think they were close to optimal conditions for ROM or just "good enough to get a signal"? How is the signal without a prepulse?

(delay, fluence, plasma scale length, optimal conditions or just good enough?, without a prepulse)

We conducted various tests with different delays and pulse energies for both the pre-pulse and the main driving pulse to optimize ROM-HHG. We did not observe ROM-HHG when the pre-pulse arrived at the target too early or late compared to the main pulse. We found that the best ROM-HHG was observed when the pre-pulse arrived at the target 2 ps earlier than the main pulse, and its intensity was approximately 10^{16} W/cm². Based on these experimental parameters, we used PIC simulations to estimate a plasma scale length of approximately 0.1λ . This plasma scale length corresponds to the scale length data in [Kahaly et al. PRL 110, 175001 (2013)].

While we have demonstrated ROM HHG using a liquid flat jet under reasonably good conditions, there are several other parameters we did not test, including focusing geometry, chirp condition, and target angle. Therefore, we cannot claim that these are the absolute best condition for ROM-HHG. However, the goal of our manuscript is not to identify the true optimal condition, but rather to present ROM-HHG using the liquid flat jet under a reasonably good condition. In response to the reviewer's suggestion, we have added the following sentences to clarify this point in the revised manuscript.

"We optimized ROM HHG by controlling the intensity and timing of the pre-pulse. We observed the best ROM HHG when the pre-pulse was arrived 2 ps earlier than the main pulse and its peak intensity was approximately 10^{16} W/cm². Under these conditions, the plasma scale length estimated by the PIC simulation was around 0.1λ ."

The main merit of the work is that it presents a clear technological breakthrough that opens many new perspectives and answers many technical questions. So for the field of ultra-intense laser-plasma interaction, and the small community working on plasma mirrors in particular, this is a major result, which I think deserves publication in a journal like Nature Communications.

We sincerely appreciate the valuable insights provided by the reviewer, whose thoughtful comments and recommendation have strengthened the quality of our manuscript and its potential for publication.

But there are also some serious technical shortcomings left that need to be addressed before I can recommend publication:

We made a point-by-point reply to the referee's comments as explained below.

Problematic comparison of experimental and PIC-simulation results in Fig. 4:

When discussing Fig.4 with the intensity dependence of the CWE harmonics, the paper states "The results of the PIC simulations, as shown in Figs. 4c and 4d, are consistent with the experimental results." This statement seems problematic to me, because the PIC simulations were made for higher intensities than the experimental ones. The simulated intensity range in Fig. 4 overlaps only for less than an order of magnitude with the experimental intensity range. So there are actually only the first 3 simulation-data-points in the experimental range, and they have a steeper intensity dependence than the linear fit. So to me the agreement is not very good.

Also to me the PIC-simulation data points in Fig.4d suggest a sub-linear intensity dependence over the simulated intensity range.

This comparison needs to be explained better, the different intensity range must be justified. The way it is presented currently is misleading because different intensity ranges are presented on top of each other as if they were the same. The different ranges must appear clearly in the graphical presentation, or at the very least be mentioned in a warning in the caption!

We completely agree with the reviewer that the different intensity ranges shown in Fig. 4 may mislead readers. However, there was a reason why we could not compare the results at the same intensity range. In the experiment, we used laser pulses with low temporal contrast. Therefore, we could not increase the laser intensity above 2×10^{16} W/cm², as we explained in the manuscript. On the other hand, we had another issue in the PIC calculation. As we reduce the laser intensity in the PIC calculation, we observed that the cutoff frequency of the CWE HHG reduces significantly. A slightly higher slope observed for the first three data points is caused by the reduction of the cutoff. Therefore, we could not compare the results at the same intensity range. Nevertheless, we agree with the necessity of a warning comment. We added the warning comment in the caption as below.

“Note that the intensity ranges of experimental results (a and b) and PIC simulations (c and d) are different because the experimental results could not be observed at high intensities due to the low temporal contrast of the laser pulse.”

I am not convinced by the photon count calibration that has been added to the paper.

- The first crucial issue is the lack of any uncertainty estimate. This must be added for both the calibration factor $c_f = 7.7$ (photons/CMOS counts) as well as for the conversion efficiency. This must account for the uncertainty of the various geometrical corrections and the precision of other assumptions that went into this calibration (thickness of oxidation layer on the filters, MCP spectral response, etc...).

The uncertainty on the photon count is caused by many different factors as the reviewer pointed out. One of them is the uncertainty of c_f that comes from the thickness estimation of the oxidation layer of the filters. We measured the transmittances of two Al filters that have different thicknesses, and concluded that the thickness of the oxidation layer is typically 40 nm (10 nm for both sides and two filters). However, even a small deviation of ± 5 nm in the thickness can result in a significant error of +47% or -32% in the transmittance T_{Al} .

We assumed errors (typically 1–10%) in other factors (C_X , C_M , q_e , and etc.). The uncertainty of c_f can be higher than 50%. In the Supplementary Information of ref. 49, it was also estimated to be 70%. As a result, the photon count presented in the manuscript is significant to estimate an order of magnitude, rather than the precise value. To avoid any potential confusion, we have included a clarification in the revised manuscript stating this point explicitly.

“The calibration factor c_f is obtained using many parameters (C_X , C_M , q_e , and etc.) that have errors. For example, the uncertainty of the transmittance estimation on the oxidation layer thickness of the filters is 40%. Therefore, the photon count presented in this work should be accepted as a representation of an order of magnitude, not an exact value.”

- I also disagree with some of the assumptions made for the estimation of the conversion efficiency on p.20:

- The total energy of the CWE beam is found by first summing up all the photons' energies that make it onto the MCP detector. Then this "observed energy" is simply scaled up by the ratio of the area of the beam cross section at the grating over the much smaller open aperture area of the grating. But that seems to implicitly assume that the average fluence outside of the observed part of the beam -notably at the edge of the 100mrad cone- is the same as in the observed part. Such a roughly homogeneous distribution, or roughly top-hat

beam, seems rather unrealistic and to me seems to lead to a clear overestimation of the conversion efficiency, doesn't it?

We appreciate the referee's careful comment. We answer this point later, together with the referee's comment starting with "The total CWE beam divergence was assumed to be..."

Shouldn't one rather consider that the grating effectively only focuses a thin 1D-slice out the polar beam distribution, and therefore one would have to integrate over θ_y , as was done in [Kahaly et al. PRL 110, 175001 (2013)] ?

We thank the referee for the correction of our equation. We integrated over $\sin\theta_y d\theta_y$, but we made a typo. We corrected the expression as shown below.

"To estimate the conversion efficiency (see Fig. S12 in the Supplementary Information), we first obtained the energy of the observed harmonics between the 7th harmonic and 14th harmonic, $E_{observed} = \iint \omega S(\omega, \theta_y) \sin\theta_y / g_e(\omega) d\omega d\theta_y$, where ω is photon energy, S is spectral intensity, θ_y is cone angle, g_e is diffraction efficiency of the grating, and the integration over θ_y was taken for the recorded range, ± 25 mrad."

- The total CWE beam divergence was assumed to be ± 100 mrad, based purely on a 2D PIC simulation. But a spectrum like that in Fig. 1e should allow an experimental estimation of the divergence, even though the detected angular part of the beam is narrow? Or can the experiments at least corroborate this very large ± 100 mrad value? The larger this assumed value is, the larger of course the extracted conversion efficiency becomes - so again I am wary of an overestimation of the conversion efficiency here. At the very least, this divergence estimate should contribute to an important uncertainty on the experimental conversion efficiency!

We agree that the angle-resolved spectrum shown in Fig. 1e shows a Gaussian-like angular distribution. On the contrary, the angular distributions of the CWE shown in Figs. 2 and S11 show uniform or double peak structures. We observed either uniform or complex distribution in most of the experiments with a narrow observable range of angular distribution, ± 25 mrad.

In the experiment, we observed 7th-10th CWE harmonics even when we controlled the angle of the liquid target by ± 175 mrad, which corresponds to the control of the alignment of the reflected beam by ± 350 mrad. The rotation of the liquid target changes the experimental condition, but it clearly indicates that the CWE beam has a large divergence

angle. We had difficulty estimating the divergence angle of the full CWE beam experimentally.

Because we could not estimate angular distributions of the full CWE beam experimentally, we had to rely on the value obtained from the PIC simulations. In the 2D PIC simulations shown in Fig. S13, 65% of the energy of the CWE beam was contained in ± 100 mrad. This choice of the value, 65% of the energy, justifies the assumption of the uniform distribution of the beam within ± 100 mrad, because the beam distributed outer than ± 100 mrad can be taken into account of the conversion efficiency by assuming uniform distribution with 95% of the peak intensity. Therefore, the estimations of the conversion efficiency obtained by Gaussian fitting and the uniform distribution with the reduced peak intensity are not much different. The error is lower than 5%.

We agree that the estimation of the angular distribution through the PIC simulation can be different from the experimental results, and we add this uncertainty to the experimental conversion efficiency.

“Considering the uncertainty of the calibration factor, grating efficiency, and the estimation of the beam divergence, the conversion efficiency presented in this work should be accepted as a representation of an order of magnitude, not an exact value.”

- The authors vaguely suggest on page 4 that only with liquid targets, kHz-rep-rate operation of plasma mirrors is possible. That is not true, as kHz-operation in the CWE regime has been demonstrated as early as 2011, and this work is cited as refs. 26, 29, 32, 34, 38. Most recently, the relativistic regime and ROM-HHG at kHz-rep-rate has also been demonstrated by the same group: [Haessler et al. *Ultrafast Science* 2022, 9893418 (2022)]. In that paper, detailed experimental parameter scans of the plasma density gradient scale length were reported. So the authors should explicitly state that kHz operation in both sub-relativistic and relativistic regimes has been demonstrated, and perfectly meaningful parameter scans are made. Once this state-of-the-art is clearly laid out, there is still a very clear case to be made why liquid sheet targets and their continuous operation for hours are far superior - be it for very extensive acquisitions like the mentioned streaking measurements, or simply for an operation of a secondary XUV source for actual applications.

We wanted to mention that the liquid target can be operated at high repetition rate continuously. We removed the incorrect statement, and added the following comment with the reference [Haessler et al. *Ultrafast Science* 2022, 9893418 (2022)] as shown below.

“Recently, HHG in the relativistic regime at a repetition rate of 1 kHz has been demonstrated using the rotating disk method³⁹ and a detailed study of the plasma scale length was reported.”

- Connected to the preceding point: in the conclusions on p.13, line 256, the authors state that the high rep-rate and long-term operation of their liquid-sheet target "allows us to analyse the detailed structure of the CWE harmonics under various conditions, such as polarisation, ellipticity, dispersion, and energy, thanks to the good stability of the light source." Their results are certainly nice, but all these studies have been made before with solid targets. So if the authors want to make such claims, they need to argue precisely what has been possible up to now, including the demonstrated 1-kHz operation with solid targets, and what will become possible only thanks to liquid-sheet targets.

We modified the following sentence to support our statement and to clarify the advantage of the liquid flat jet.

“Thanks to the continuous operation of the liquid flat-sheet plasma mirror at high repetition rates, we were able to observe the structure of the CWE harmonics for various laser parameters, including polarization, ellipticity, dispersion, and energy, all in a single day. This achievement would have been challenging if a target supports a limited number of laser shots.”

- The authors vaguely state (on p.4 line 71) that rotating disks and tape targets "suffer from mechanical instability at high-repetition-rate operations". This is a "null statement", because it remains open what "instability" is suppose to mean. What stability requirement for which application is not reached with solid targets? The authors cite ref. 34 where active stabilization (albeit for a ridiculously short amount of time) has been shown to enable interferometric stability, i.e. ~100nm position stability of a rotating disk. It is not clear that a liquid sheet target could do any better. So I don't think the authors can so easily claim, without hard evidence, that a liquid sheet target would solve any stability issues.

- A similar vague reference to "stability" is made when discussing Fig. 3b (p.9, line 186), when it is state that it "showed the detailed structure of the spectra depending on the GDD values owing to the good stability of the light source". Again: what precisley is meant by that? What kind of instability would blur the spectral structures?

We agree that using “stability” or “instability” in our manuscript is vague without enough supporting data. We revised the manuscript as follows.

“However, these solutions ~~suffer from mechanical instability at high-repetition-rate operations~~^{32,34} ~~and still~~ provide a limited number of laser shots.”

“The high-flux HHG from the liquid plasma mirror allowed us to observe the CWE dynamics with good ~~stability-statistics~~ (see Figs. S5 and S6 for the ~~stability measurement in Supplementary Information~~).”

“**Fig. S5. ~~Stability-Statistics~~ of 1-kHz CWE harmonic over 22 minutes.** **a**, CWE spectra (1357 spectra) taken with an exposure time of 101 ms. The color code shows the number of the spectrum overlapped. The time interval between each spectrum was around 400 ms to save the spectrum in a computer. **b**, Beam pointing ~~stability-statistics~~. The data points denote the position of the harmonic spectrum (θ_y) for the spectral range from $6.5\omega_0$ to $14.5\omega_0$. The solid red line shows smoothed data points. ”

“**Fig. S6. ~~Stability-Statistics~~ of CWE harmonics obtained from a liquid plasma mirror recorded in a single-shot-based experiment.** **a**, Spectral ~~stability-statistics~~ of the single-shot CWE spectra obtained from a liquid plasma mirror. The solid gray lines are 120 single-shot spectra obtained in 2 minutes. The solid red line is the mean value of the 120 spectra. **b**, Beam pointing ~~stability-statistics~~ of the CWE harmonic beam obtained from a liquid plasma mirror. Each data point corresponds to the angle where the maximum intensity is observed.”

Minor remarks:

- "Liquid plasma mirror" in the title seems a bit too general, given that there has been one experiment with a liquid jet before [Heissler et al. New Journal of Physics 16, 113045 (2014)]. I suggest specifying "flat liquid-sheet" or so.

We agree with the reviewer that the title was too general. We changed the title as below.

“High-harmonic Generation from a Flat Liquid-sheet Plasma Mirror”

- The formulation "Its usage is limited to applications such as time-resolved studies and attosecond streaking experiments..." is misleading, because this passive form means that "only such applications are possible". The contrary is true: no such experiments could be done with plasma mirror harmonics so far. So I suppose the authors mean that its usage limits, or even prevents such applications. So: the active form "limits"/"prevents" should be

used rather than the passive "is limited to".

We thank the referee for the kind correction of the sentence. We revised the sentence as follows.

“Thus, it is challenging to use a solid target for applications such as time-resolved studies and attosecond streaking experiments that require a large amount of data.”

- On p.10, line 197, the sentence "The spectral intensity was estimated from the sum of the spectrum from $6.5\omega_L$ to $14.5\omega_L$." is not clear to me. If one "sums a spectrum" along its spectral dimension, one no longer has "spectral" intensity, right? I suppose this refers to the ordinate axis of Fig. 4b, which shows photon number/shot, i.e. not at a "spectral intensity" but something corresponding to a pulse energy. The description of panel 4b as "the sum of the peak intensity of the CWE spectra (b) between $6.5\omega_L$ to $14.5\omega_L$ " given in the caption of Fig.4 is equally incomprehensible to me, and I am left guessing what the connection to the photons/shot may be. This must be cleared up.

We thank the referee for the careful corrections of the misleading sentence. We modified the sentences following your kind suggestions as follow.

“The **observed photon number per laser shot** of the harmonic radiation is shown in Fig. 4b as a function of the peak intensity. The **observed photon number per laser shot** was estimated from the sum of the spectrum from $6.5\omega_L$ to $14.5\omega_L$. The CWE **photon number** recorded in the experiment exhibited a linear dependence (Fig. 4b, red dashed line, $I_{CWE-EXP} \propto I_0^{1.1}$).”

“**Fig. 4. Intensity scaling of CWE from a liquid plasma mirror. a,b**, Angle-integrated CWE spectra (**a**) and the sum of the **observed spectral intensity** of the CWE spectra (**b**) between $6.5\omega_0$ and $14.5\omega_0$ obtained in experiments for different peak intensities of the driving laser. The data points are the mean values of 5 independent measurements. The error bars show the standard deviation. **c,d**, Angle-integrated CWE spectra (**c**) and the sum of the intensity of the CWE spectra (**d**) between $6.5\omega_0$ and $14.5\omega_0$ obtained in 2D PIC simulations obtained at the different peak intensities of the laser pulse. **Note that the intensity ranges of experimental results (a and b) and PIC simulations (c and d) are different because the experimental results could not be observed at high intensities due to the low temporal contrast of the laser pulse.**”

- On p.11, line 225, "the ROM model" is mentioned, without a reference. The authors should not suggest that there is a canonical "ROM model", because that is untrue. There has been a period of time when the theory of ref.17 was considered to provide a "universal spectral shape" of $q^{-8/3}$ (used to fit the exp. ROM spectrum in Fig.5), and a fundamental understanding. Today, our understanding has much advanced, problematic assumption in the Baeva-model are recognized, and we know about a greater complexity of the involved relativistic plasma dynamics. Much less steep spectral shapes $q^{-4/3}$ are predicted, e.g. in refs. 21 and 67.

We agree with the reviewer that the Baeva's ROM model could not be universally applied. Therefore, the guide line drawn with $q^{-8/3}$ would be inappropriate. In fact, it has little meaning to fit the spectral intensity unless the spectral responses of the grating and the MCP are not accurately calibrated. Therefore, we removed the guideline drawing in Fig. 5. We also revised the manuscript as below.

"The peak intensity of the laser pulse was estimated to be 1.7×10^{20} W/cm², which was considerably higher than the relativistic intensity. ~~The spectral intensity of the ROM harmonics fits well with the ROM model (Fig. 5).~~ Thus, we conclude that the observed spectrum was generated in the ROM regime."

Fig. 5. High-harmonic spectrum generated in the ROM regime using a liquid plasma mirror. The upper panel shows the angle-resolved spectrum, and the lower panel shows the angle-integrated HHG spectrum generated through ROM from a liquid plasma mirror. The black arrows denote the second-order diffraction of the harmonics. ~~The orange dashed line is a $q^{-8/3}$ fitting line to the spectrum, where q is the photon energy of the harmonics.~~

- On p.13, line 265, the authors write "We also showed that the harmonic intensity increases linearly as the peak intensity of the driving laser pulse increases." This must refer to Fig.4 with CWE results, but starts the paragraph about the ROM-results, which is confusing. Also, this mention suggests that this linear scaling (i.e. constant conversion efficiency) would be a new results, which it is not: see Fig. 1 in [Quéré et al. Physical Review Letters 96, 125004 (2006)].

We agree that the sentence is located in an inappropriate position. We removed the sentence as below.

~~"We showed that the harmonic intensity increases linearly as the peak intensity of the driving laser pulse increases. HHG from the liquid plasma mirror in the ROM regime was demonstrated using a 150-TW-laser with a pulse energy of 5 J. We demonstrated HHG from the liquid plasma mirror in the ROM regime using a 150-TW-laser with a pulse energy of 5 J. According to the existing research^{17,21,68}, the conversion efficiency of HHG from a plasma mirror varies depending on the harmonic order q as q^{-p} with the exponent $p = 4/3 \sim 8/3$. Thus, attosecond pulses can be obtained with energies of ~mJ or even higher. These estimations indicate that the liquid plasma mirror is a promising light source for the generation of intense, high-repetition-rate attosecond pulses. Consequently, the results of this study will significantly extend the applicable areas of HHG from a plasma mirror, including time-resolved pump-probe studies and attosecond streaking experiments in both the linear and nonlinear regimes."~~

- At the end of the conclusions, on page 14, ROM-HHG with a kHz-driver is mainly presented as an outlook, and new kHz-lasers with tens of mJ pulse energies are mentioned. But the state of the art includes ROM at 1kHz rep-rate with few-mJ drivign pulses, as demonstrated in [Haessler et al. Ultrafast Science 2022, 9893418 (2022)].

We thank the referee for the kind correction. We revised the sentence as shown below.

~~"Thus, these laser systems can provide a laser pulse with a high peak intensity on the target that can reach up to $\sim 10^{20}$ W/cm², which is sufficient for the generation of intense ROM harmonics."~~

- In the description of the PIC simulations on p.21, the authors mention the plasma density scale length of 0.006λ . This is an extremely short value - how was this chosen? CWE is typically said to be optimized for a $\lambda/100$ to $\lambda/50$ scale length (see e.g. ref.3)- so 2 or 3 times longer.

It is clearly unrealistically short for an experiment. Even if the peak intensity was only $1e16$ W/cm², the contrast ratio was also only $1e6$. Have the authors done simulations with other longer scale lengths? How robust was the agreement of the linear-polarization scan (Fig.2) with the experiment for different simulated conditions?

We performed many PIC simulations with different plasma scale lengths from 0.001λ to 0.1λ to find the optimal scale length for CWE HHG. We chose the optimal condition by comparing the 6th–14th harmonics intensity, and it was optimal at the plasma scale length of 0.006λ . We would like to mention that it is not unrealistically short. For example, in the previous work [Kahaly et al. PRL 110, 175001 (2013)], it is shown in FIG. 3a that the plasma scale length of 0.008λ and the maximum electron density $200n_c$ were used for CWE HHG. These numbers are very close to what we used for our 2D PIC simulations. Therefore, based on our PIC simulations and previous work, we expect that the linear polarization angle dependence of CWE would not be too sensitive to the plasma scale length.

Why are the harmonics peaks in the simulations in Fig.3 so much more narrow than in the experiments? Maybe because the gradient scale length is longer in the experiments?

This is probably due to the spectral resolution of our EUV spectrometer. We did not use a slit in the EUV spectrometer. The opening width of the grating (Shimadzu, 30-006 and 30-007) plays a role of a slit, which was 2.6 mm. Although the resolution of the spectrometer was not the best, it was good enough to distinguish each harmonic peak and observe GDD dependence.

- In the caption of Fig.5, "5 independent experiments" are mentioned. What exactly means 'independent'? 5 successive acquisitions? Or from 5 different days with their individual alignment etc.? How does a scan like Fig; 3a look without such averaging?

In the caption of Fig. 5, we did not mention 5 independent experiments. In the caption of Figs. 2-4, we mentioned "5 independent experiments" or "5 independent measurements". We meant that we made 5 successive scans without changing experimental conditions. In Fig 3a, the experimental data were taken for 5 successive scans, and their mean values are shown. Since it took a long time (10 min for 5 scans), the experimental condition could be changed slightly. Therefore, it looks like a data obtained without averaging.

REVIEWERS' COMMENTS

Reviewer #4 (Remarks to the Author):

The authors took into account all my comments and replied well to my criticism. Unclear points have been cleared up and I think the paper is now ready to be published in Nature Communications. Congratulations to this very impressive work!

I have one tiny remark:

when describing the prepulse properties used for the ROM experiments, its delay and intensity are given, but also the *fluence* (in J/cm^2) would be important to know since this is what mainly scales the plasma-expansion velocity.

Response to referees

As the reviewer suggested adding the fluence information, we have modified the manuscript as written below.

"We observed the best ROM HHG when the pre-pulse arrived 2 ps earlier than the main pulse. The peak intensity and fluence of the pre-pulse were approximately 10^{16} W/cm² and 800 J/cm²."